# Research on a Simulation Method of the Millimeter Wave Radar Virtual Test Environment for Intelligent Driving

**DOI:** 10.3390/s20071929

**Published:** 2020-03-30

**Authors:** Xin Li, Xiaowen Tao, Bing Zhu, Weiwen Deng

**Affiliations:** 1State Key Laboratory of Automotive Simulation and Control, Jilin University, Changchun 130022, China; 2Aviation University of AF, Changchun 130022, China; 3School of Transportation Science & Engineering, Beihang University, Beijing 100191, China

**Keywords:** intelligent driving, virtual test environment, millimeter wave radar

## Abstract

This study addresses the virtual testing of intelligent driving, examines the key problems in modeling and simulating millimeter wave radar environmental clutter, and proposes a modeling and simulation method for the environmental clutter of millimeter wave radar in intelligent driving. First, based on the attributes of intelligent vehicle millimeter wave radar, the classification characteristics of the traffic environment of an intelligent vehicle and the generation mechanism of radar environmental clutter are analyzed. Next, the statistical distribution characteristics of the clutter amplitude, the distribution characteristics of the power spectrum, and the electromagnetic dielectric characteristics are analyzed. The simulation method of radar clutter under environmental conditions such as road surface, rainfall, snowfall, and fog are deduced and designed. Finally, experimental comparison results are utilized to validate the model and simulation method.

## 1. Introduction

Compared to traditional driving, intelligent driving can effectively solve issues such as human and vehicle safety and shared travel and has become a focus of research in the automotive industry and vehicle engineering. Intelligent driving has also become a core area of competition for high-tech enterprises working with artificial intelligence and the Internet, and the world’s technological powers have incorporated intelligent driving in their science and technology development plans [1]. In the research and development of intelligent driving, virtual simulation tests can bypass bottlenecks such as the long cycle, high cost, and low safety of actual vehicle tests. Repeated testing in complex traffic scenarios is necessary for intelligent vehicles to be accepted by the public and ultimately to be safe on the road [2,3]. Millimeter wave radar has the advantages of technological maturity, wide application, low cost, high precision, and good stability in the traffic environment, and forms the basis of the indispensable sensors used in intelligent driving [1,4].

In the millimeter wave radar virtual testing of intelligent driving at both domestic and foreign sites, the simulation model of the millimeter wave radar test environment usually does not consider the mechanisms responsible for the generation of radar clutter, and hence the radar environmental clutter cannot change dynamically with the traffic scene. As a result, the simulation results of millimeter wave radar intelligent driving tend to be idealized, which does not objectively reflect the actual radar detection mechanism. This is an important issue that urgently needs to be solved in the virtual testing of intelligent driving.

To perform such virtual testing requires a complete set of high-fidelity millimeter wave radar system simulation model inputs. Among them, radar environmental clutter is the key factor affecting radar detection and measurement. Strong clutter background can lead to problems such as radar false alarms, missed detection, and measurement error [1,5,6]. Ignoring environmental clutter in the modeling and simulation research of millimeter wave radar will greatly reduce the fidelity of the model, which will seriously affect the credibility of virtual testing. This study addresses intelligent driving virtual tests in terms of the modeling and simulation of millimeter wave radar environmental clutter. This investigation has significant advantages, which can effectively solve the problem that the existing radar virtual test environment cannot change dynamically with the traffic scene, which results in a tendency for the radar simulation results to be idealized.

## 2. Analysis of Environmental Clutter Mechanism

Millimeter wave radar works in actual traffic scenes. In addition to radar-focused targets such as vehicles and pedestrians, there are non-radar target elements such as roads, buildings, transportation facilities, and weather conditions. These elements reflect radar electromagnetic waves, and these reflections are collectively referred to as radar clutter. The environmental clutter of the millimeter wave radar of intelligent vehicles consists mainly of ground clutter and weather clutter.

### 2.1. Analysis of Ground Clutter

The surface traffic depicted in Figure 1 generates ground clutter, which is the distributed scattering echo of the incident electromagnetic wave of the radar, a phenomenon that exerts a great influence on intelligent driving millimeter wave radar. In general, ground clutter is extremely unstable. For example, wind causes micro-motions of objects such as trees and grass, which can cause amplitude fluctuations and spectral broadening of ground clutter [7,8].

When examining the ground clutter mechanism, we should focus on the amplitude and frequency domain characteristics of ground clutter, which are affected by factors such as the wavelength of incident electromagnetic waves, surface area of radar radiation, incident angle of radar, polarization mode of incident electromagnetic waves, complex dielectric properties of the surface, and ground roughness [9,10,11].

In the study of the amplitude characteristics of ground clutter, it is usually necessary to solve the probability distribution of the amplitude based on the actual surface characteristics. Common amplitude distributions that can be used for ground clutter include the Lognormal, Weibull, and K distributions. The probability density distribution of the radar clutter amplitude describes the amplitude characteristics of the clutter signal in the time domain. To better describe the distribution characteristics of ground clutter generally requires an analysis of its spectral distribution characteristics. Common spectral distributions include the Gaussian, Cauchy, and Omnipolar distributions [10,12]. The distribution should be determined by fitting the scene surface data.

### 2.2. Analysis of Weather Clutter

Since the operating wavelength of millimeter wave radar for intelligent driving is on the order of millimeters, the wavelength is similar to the diameter of meteorological particles such as rain and snow. According to theory, the phase change of the incident field along the target length is more significant when the wavelength of the incident electromagnetic wave and the target size are of the same order of magnitude. Within the scattering region, each part of the weather scatterer affects the other parts [7,13,14]. The field strength at each point on the scatterer is the superposition of the scattered field strengths caused by the incident point and the remaining points in the scatterer, and the total effect of the interactions between the various parts of the scatterer determines the density distribution of the final current. For this scattering method, the exact Stratton–Zhulan integral equation must be solved in order to obtain the scattering field solution, usually with the moment solution [15,16,17].

The more common weather scenes of intelligent vehicles include rainy days, snowy days, and foggy days, as shown in Figure 2.

For weather scenes such as rain, snow, and fog, during radar detection there are many weather particles in each radar resolution unit. Assuming that the radar cross-section (RCS) of each weather particle is $\sigma_{i}$, the total RCS of the weather in the radar resolution unit is the sum of the RCS of all weather particles [10,12],
(1)$$\sigma_{c}=V_{c}G_{\eta }=V_{c}\sum_{i}\sigma_{i},$$
where $G_{\eta }$ is the radar cross-sectional area of the weather particles per unit volume, and $V_{c}$ is the total volume of the radar spatial resolution unit, which can be expressed as
(2)$$V_{c}=\frac{\pi }{4}\left(R\theta_{B}\right)\left(R\phi_{B}\right)\left(\frac{cT}{2}\right)\frac{1}{2ln2},$$
where *R* is the radial distance corresponding to the radar spatial resolution unit; T is the single-frequency modulation time of the radar; $\theta_{B}$ and $\phi_{B}$ are, respectively, the horizontal and vertical half-power beam widths of the radar antenna; and *c* is the propagation speed of the radar wave. If the diameter of the weather particle is $D_{i}$, then the cross-sectional area of the radar can be expressed as
(3)$$\sigma_{i}=\frac{\pi^{5}D^{5}{}_{i}}{\lambda^{5}}{\left|K\right|}^{2},$$
where ${\left|K\right|}^{2}=\left(\varepsilon -1\right)/\left(\varepsilon +2\right)$, and ε is the dielectric constant of the weather particles. Since ε is temperature dependent, ${\left|K\right|}^{2}$ changes with temperature.

The radar cross-sectional area η of weather particles per unit volume is
(4)$$\eta =\sum_{i}\sigma_{i}=Tf^{4}r^{1.6}\times 10^{-12}\;m^{2}/m^{3},$$

The radar cross-sectional area of the weather particles is proportional to the 1.6th power of its diameter $D_{i}$. Let $\sum_{i}D_{i}$ be the radar reflectivity factor of the weather particles, denoted by *Z*. Its mathematical description varies with rain, snow, fog, and other weather conditions. The radar reflectivity factor of rain is
(5)$$Z_{rain}\approx 200r^{1.6},$$

For dry snowfall, snow particles are mainly composed of ice crystals, single crystals, or synthetic crystals. The radar reflectivity factor of snow is
(6)$$Z_{snow}\approx 1780r^{2.21},$$

Mist is a floating combination of tiny water droplets or ice crystals close to the ground. Inland fog is usually radiation fog, whose average droplet diameter is generally < 20 μm [14]. The radar reflectivity factor of fog is
(7)$$Z_{frog}=4.62\times 10^{-4.16}{\left(V\right)}^{-3.16},$$
where r is the rainfall or snowfall rate in mm/h and *V* is the visibility of the fog in m.

## 3. Modeling of Environmental Clutter

### 3.1. Modeling of Ground Clutter

Ground clutter can be seen as a multi-point scattering set on the road surface, in which there is serious random mutual interference between the scattered signals. Ground clutter is a non-stationary random signal that changes with time.

The modeling of ground clutter must consider the strong scattering echoes of both ground clutter and ground stationary objects. The architecture of ground clutter modeling is shown in Figure 3.

In the study of ground clutter modeling, as shown in Figure 4, we distinguish three types of road environments: highways, urban roads, and rural roads.

A large number of ground clutter data measurements were carried out using self-developed millimeter-wave raw data radar.

According to the analysis of the ground clutter mechanism, the usual power spectrum distributions of ground clutter include the Gaussian, Cauchy, and Omnipolar distributions. For intelligent driving millimeter wave radar, we used the statistical analysis toolbox of MATLAB 2019 software. The ground clutter statistical fitting analysis of intelligent driving millimeter wave radar was performed using the measurement data from Delphi radar, TRW radar, and Continental radar in the multi-traffic road scene, as shown in Figure 5. The data used in the fitting analysis were from 266 actual road tests of the millimeter wave radar conducted from 2017–2019, and the test scenes consisted of highways, urban roads, and rural roads.

Using the abundant statistical analysis function library of MATLAB software, the Gaussian, Cauchy, and Omnipolar distributions were fitted in turn, and the data fitting results were analyzed synthetically. From the statistical results, we deduced that the ground clutter power spectrum of the intelligent driving millimeter wave radar in three types of traffic scenes most closely obeys a Gaussian distribution. Therefore, we utilized the Gaussian distribution power spectrum function to model the ground clutter of the intelligent driving millimeter wave radar.

The power spectrum of the Gaussian distribution is shown in Figure 6 and can be described mathematically as
(8)$$S\left(f\right)=exp(-\frac{{\left(f-f_{d}\right)}^{2}}{2\sigma_{f}^{2}}),$$
where $f_{d}$ is the average Doppler shift of the clutter power spectrum, and $\sigma_{f}$ is its standard deviation. $f_{d}$ is mainly affected by the speed of the vehicle on which the radar is mounted, i.e., $f_{d}=2v_{c}/\lambda$, where $v_{c}$ is the speed of the radar’s vehicle, and λ is the working wavelength of the radar.

The commonly used statistical distributions of the amplitude probability density of ground clutter include Lognormal, Weibull, and K distributions. Similarly, using the statistical analysis toolbox of MATLAB, the amplitude probability density distributions commonly used in ground clutter were fitted with the measurement data of the multi-radar in multi-class road traffic scenes. From a comprehensive analysis of the fitting results, we concluded that the probability density distribution of the ground clutter amplitude of intelligent driving millimeter wave radar in traffic scenes such as highways, urban roads, and rural roads is more similar to the Weibull distribution. Therefore, we used the amplitude probability density of the Weibull distribution to model the ground clutter of the intelligent driving millimeter wave radar.

The Weibull distribution of *x*, which is the amplitude of the clutter echo, is
(9)$$f\left(x\right)=\frac{px^{p-1}}{q}exp[-{\left(x/q\right)}^{p}],$$
where p is related to factors such as the degree of undulation and continuity of the road surface, and q is related to the reflection intensity and echo power of the ground clutter.

For highways, urban roads, and rural roads, we carried out statistical analysis and fitting experiments for the ground clutter data of 30 different traffic scenes using MATLAB software. We substituted the measured ground clutter data into the Weibull distribution function in order to solve the parameters p and q and then averaged the parameter values obtained from 30 parameter solution processes in order to obtain the recommended values of p and q in the three traffic road scenes, as shown in Figure 7, Figure 8 and Figure 9. In these figures, the blue dots represent the solution values of p and q in each experiment, and the red dotted lines represent the estimated average values of p and q, which were used as the recommended values for p and q.

The estimated average values of parameters p and q for highways, urban roads, and rural roads are listed in Table 1.

In addition, we compared and analyzed the statistical error between the actual road clutter data and the ground clutter simulation data, which obeys the power spectrum of the Gaussian distribution and the amplitude probability density of the Weibull distribution. After randomly selecting 90 sets of actual road clutter data, each with a time period of 5 s, we calculated the average amplitude of the clutter time domain signal for each time period. At the same time, we utilized the above statistical distribution characteristics to generate simulated clutter data and to calculate the average amplitude of the simulation clutter data. On this basis, we normalized the amplitude data, and the error between the actual ground clutter and the simulated ground clutter was calculated. The error statistics comparison results are presented in Figure 10. From this comparison, we can see that the error between the actual clutter and the simulated clutter is smaller and the consistency is better.

Considering the echo of the ground stationary to be strongly scattered, and superimposing it on the ground clutter, the simulated flow of the obtained ground clutter is shown in Figure 11.

Let w=u+jv be a complex variable of a Weibull distribution, which can be generated by the transformation of the complex Gaussian random variable m=x+jy:(10)$$\{\begin{array}{l} u=\frac{x{\left(x^{2}+y^{2}\right)}^{\frac{1}{p}}}{\sqrt{\left(x^{2}+y^{2}\right)}}\\ v=\frac{y{\left(x^{2}+y^{2}\right)}^{\frac{1}{p}}}{\sqrt{\left(x^{2}+y^{2}\right)}} \end{array},$$
where x and y are Gaussian variables distributed as $N\left(0,\sigma^{2}\right)$ ($\sigma^{2}=q^{p}/2$).

In Figure 11, the strong ground scattering facility is an arbitrary strong scattering target that is stationary relative to the road surface. The radar echo is
(11)$$s_{r}\left(t\right)=e^{j2\pi \left[f_{0}\left(t-\frac{2R}{c}\right)+\pi {\left(t-\frac{2R}{c}\right)}^{2}/\lambda \right]},$$
where $f_{0}$ is the radar carrier frequency, λ is the working wavelength of the radar, and R is the distance from the stationary target to the radar.

For example, in the traffic scene of a city road, let p=7,q=6. When two stationary parked vehicles are added to the simulation, the time domain signal of the complex Weibull clutter generated by the above modeling method is the result shown in Figure 12.

The comparison between the complex Weibull clutter generated by the ground clutter modeling method and the ideal Weibull curve is shown in Figure 13. It can be seen from this figure that the curve fitting effect of the model is better, and the injection effect of the strong scattering target of the two vehicles is also significant.

### 3.2. Modeling of Weather Clutter

Considering the rainfall attenuation rate, reflectivity, amplitude, and phase distribution of factors such as rain clutter and radar transmission power, the rain clutter modeling method was designed, as shown in Figure 14.

Based on the analysis of the rain clutter mechanism, combined with the physical characteristics of the rain in a traffic scene, the motion characteristics, and the millimeter wave radar characteristics, the simulation method for rain clutter in the time domain signal was constructed. The size distribution of raindrops varies with the type of rainfall, i.e., light, moderate, or heavy rain. The raindrop size distribution of each type of rainfall is
(12)$$N\left(d_{min}\right)=\{\begin{array}{c} 60000e^{-5.7R_{rain}{}^{-0.21}d_{rain}}\;\mathrm{light}\;\mathrm{rain}\\ 14000e^{-41R_{rain}{}^{-0.21}d_{rain}}\;\mathrm{moderate}\;\mathrm{rain}\\ 2800e^{-3.0R_{rain}{}^{-0.21}d_{rain}}\;\mathrm{heavy}\;\mathrm{rain}, \end{array}$$
where $d_{rain}$ is the raindrop diameter in mm, and $R_{rain}$ is the rainfall rate in mm/h. The dielectric constant of rainwater can be described by a complex number whose real and imaginary parts are
(13)$$\varepsilon_{rain1}=\varepsilon_{\infty }+\frac{\left(\varepsilon_{x}-\varepsilon_{\infty }\right){\left(1+\frac{\lambda_{s}}{\lambda }\right)}^{1-\alpha }\sin\left(\alpha \frac{\pi }{2}\right)}{1+2{\left(\lambda_{s}/\lambda \right)}^{1-\alpha }\sin\left(\alpha \frac{\pi }{2}\right)+{\left(\lambda_{s}/\lambda \right)}^{2\left(1-\alpha \right)}},$$
(14)$$\varepsilon_{rain2}=\frac{\sigma \lambda }{18.8496\times 10^{10}}+\frac{\left(\varepsilon_{x}-\varepsilon_{\infty }\right){\left(\frac{\lambda_{x}}{\lambda }\right)}^{1-\alpha }\cos\left(\alpha \frac{\pi }{2}\right)}{1+2{\left(\lambda_{s}/\lambda \right)}^{1-\alpha }\sin\left(\alpha \frac{\pi }{2}\right)+{\left(\lambda_{s}/\lambda \right)}^{2\left(1-\alpha \right)}}.$$

Let T (°C) be the ambient temperature. The parameters in the above equations are
(15)$$\varepsilon_{x}=78\left[1-4.6\times 10^{-3}\left(T-25\right)+1.2\times 10^{-5}{\left(T-25\right)}^{2}-2.8\times 10^{-8}{\left(T-25\right)}^{3}\right],$$
(16)$$\varepsilon_{\infty }=5.3+2.2\times 10^{-2}T-1.3\times 10^{-3}T^{2},$$
(17)$$\lambda_{s}=3.4\times 10^{-4}e^{\left(\frac{2513}{T+273}\right)},$$
(18)$$\alpha =-\frac{16.8}{T+273}+6.1\times 10^{-2},$$
(19)$$\sigma =12.6\times 10^{8}.$$

Using Mie scattering theory, the analytical solution of rain particle scattering was obtained by solving Maxwell’s equations of wavelength-sized particles. The radar cross-section of the total rainfall attenuation is
(20)$$\sigma_{t}\left(d_{rain}\right)=\frac{2\pi c^{2}}{f_{c}{}^{2}}\sum_{n=1}^{\infty }\left(2n+1\right)Re\left(a_{n}+b_{n}\right),$$
and the backscattering segment of the rainfall is
(21)$$\sigma_{b}\left(d_{rain}\right)=\frac{\pi c^{2}}{f_{c}{}^{2}}{\left|\sum_{n=1}^{\infty }\left(2n+1\right){\left(-1\right)}^{n}\left(a_{n}-b_{n}\right)\right|}^{2},$$
where
(22)$$a_{n}=\frac{\psi_{n}(\alpha )\psi_{n}^{\prime }(\beta )-m\psi_{n}(\beta )\psi_{n}^{\prime }(\alpha )}{\xi_{n}(\alpha )\psi_{n}^{\prime }(\beta )-m\psi_{n}(\beta )\xi_{n}^{\prime }(\alpha )},$$
(23)$$b_{n}=\frac{m\psi_{n}(\alpha )\psi_{n}^{\prime }(\beta )-\psi_{n}(\beta )\psi_{n}^{\prime }(\alpha )}{m\xi_{n}(\alpha )\psi_{n}^{\prime }(\beta )-\psi_{n}(\beta )\xi_{n}^{\prime }(\alpha )}.$$

Here, α and β are related to the dielectric constant of rain, the size of the raindrop particles, the carrier frequency of the radar wave, and the propagation speed. The quantitative relationships are
(24)$$\alpha =\frac{f_{c}}{c}d_{rain},$$
(25)$$\beta =\frac{f_{c}}{c}d_{rain}\sqrt[{4}]{\varepsilon^{2}{}_{rain1}+\varepsilon^{2}{}_{rain2}}$$

Let us set $\psi_{n}(x)$ as the first type of Bessel function and $\zeta_{n}(x)$ as the second type of Hankel function,
(26)$$\psi_{n}(x)=xj_{n}(x)=\sqrt{\pi x/2}J_{n+\frac{1}{2}}(x),$$
(27)$$\xi_{n}(x)=xh_{n}^{\left(1\right)}(x)=\sqrt{\pi x/2}H_{n+\frac{1}{2}}^{\left(1\right)}(x),$$

We find that the respective decay rate and reflectance of rainfall are
(28)$$\gamma_{rain}=10^{\left(4.343\times 10^{2}\int \sigma_{t}\left(d_{rain}\right)N\left(d_{rain}\right)dd_{rain}\right)-3},$$
(29)$$\eta_{rain}=\int \sigma_{b}\left(d_{rain}\right)N\left(d_{rain}\right)dd_{rain}.$$

The amplitude variation of rain clutter obeys the Rayleigh distribution, and the phase change is uniformly distributed. When there is wind in the ambient environment, the clutter spectrum becomes $f_{d}=f_{w}+f_{0}$, where $f_{w}$ and $f_{0}$ are the Doppler shifts of the wind speed and radar, respectively.

Similarly, considering the snow attenuation rate, reflectivity, amplitude and phase distribution of snow clutter, and radar transmission power, the modeling of snow clutter was determined.

Our analysis revealed that snowfall is more sensitive to the influence of ambient temperature. At the same time, the correlation between light snow, moderate snow, and heavy snow is relatively high. Therefore, we no longer classify the snowfall based on snowfall intensity, but according to the temperature of the environment. The size distribution of the snowfall particles with diameter $d_{snow}$ is
(30)$$N_{s}\left(d_{snow}\right)=\{\begin{array}{cc} 2.5\times 10^{3}R_{snow}{}^{-0.94}d_{s}^{1/3}e^{-2.29R_{snow}{}^{-0.45}d_{snow}^{4/3}} & T\leq -10\;{}^{{}^{\circ}}C\\ 9.25\times 10^{2}R_{snow}{}^{-0.94}d_{s}^{1/3}e^{-2.29R_{snow}{}^{-0.45}d_{snow}^{4/3}} & T>-10\;{}^{{}^{\circ}}C \end{array},$$
where $d_{snow}$ is the snow particle diameter in mm, and $R_{snow}$ is the snowfall rate in mm/h. The calculation of the complex permittivity of snowflake particles is similar to that of rain particles, but the real and imaginary parts are different:(31)$$\varepsilon_{snow1}=\varepsilon_{\infty }+\frac{\left(\varepsilon_{x}-\varepsilon_{\infty }\right){\left(1+\frac{\lambda_{s}}{\lambda }\right)}^{1-\alpha }sin\left(\alpha \frac{\pi }{2}\right)}{1+2{\left(\lambda_{s}/\lambda \right)}^{1-\alpha }sin\left(\alpha \frac{\pi }{2}\right)+{\left(\lambda_{s}/\lambda \right)}^{2\left(1-\alpha \right)}},$$
(32)$$\varepsilon_{snow2}=\frac{\sigma \lambda }{19\times 10^{10}}+\frac{\left(\varepsilon_{x}-\varepsilon_{\infty }\right){\left(\frac{\lambda_{x}}{\lambda }\right)}^{1-\alpha }cos\left(\alpha \frac{\pi }{2}\right)}{1+2{\left(\lambda_{s}/\lambda \right)}^{1-\alpha }sin\left(\alpha \frac{\pi }{2}\right)+{\left(\lambda_{s}/\lambda \right)}^{2\left(1-\alpha \right)}},$$
where
(33)$$\varepsilon_{x}=203+2.5T+0.15T^{2},$$
(34)$$\varepsilon_{\infty }=3.168,$$
(35)$$\alpha =0.288+0.0052T+0.00023T^{2},$$
(36)$$\lambda_{s}=10^{-4}e^{\left(\frac{1320000}{2\left(T+273\right)}\right)},$$
(37)$$\sigma =1.26e^{\left\{-12500\left(T+273\right)\right\}}.$$

Mist consists of tiny water droplets or ice crystals. Fog droplets near the surface of the Earth are usually meteorological particles with an average diameter < 20. Considering the transmission power of the intelligent vehicle millimeter wave radar, the attenuation rate of fog, reflectivity, amplitude of the fog clutter, and phase distribution, the modeling method of the fog clutter was determined.

The size distribution of the droplets is highly correlated with visibility and is described as
(38)$$N\left(d_{fog}\right)=\frac{9.8}{{\left(42000Y_{fog}\right)}^{-1.7}}\cdot 10^{9}\cdot \frac{d_{fog}}{2}e^{\left(-\frac{6.25}{{\left(42000Y_{fog}\right)}^{-0.5}}d_{fog}\right)},$$
where $d_{fog}$ is the raindrop diameter in mm, and $Y_{fog}$ is the fog visibility in m. The real and imaginary parts of the dielectric constant of the mist particle are calculated respectively as
(39)$$\varepsilon_{fog1}=\frac{\varepsilon_{x}-\varepsilon_{a}}{1+{\left(c/\lambda f_{p}\right)}^{2}}+\frac{\varepsilon_{a}-\varepsilon_{b}}{1+{\left(c/\lambda f_{s}\right)}^{2}}+\varepsilon_{b},$$
(40)$$\varepsilon_{fog2}=\frac{c\left(\varepsilon_{x}-\varepsilon_{a}\right)}{\lambda f_{p}\left[1+{\left(c/\lambda f_{p}\right)}^{2}\right]}+\frac{c\left(\varepsilon_{a}-\varepsilon_{b}\right)}{\lambda f_{s}\left[1+{\left(c/\lambda f_{s}\right)}^{2}\right]},$$
where
(41)$$\varepsilon_{x}=77+103\left(\theta -1\right),$$
(42)$$\varepsilon_{a}=5.48,\varepsilon_{b}=3.51,$$
(43)$$f_{p}=20-142\left(\theta -1\right)+294{\left(\theta -1\right)}^{2},$$
(44)$$f_{s}=590-1500(\theta -1),$$
(45)θ=300/T

## 4. Simulation Application of Environmental Clutter

### 4.1. Simulation Application of Ground Debris Wave

In order to verify the effectiveness of the ground clutter modeling method, we set up a traffic simulation scenario to simulate the application of ground clutter.

#### 4.1.1. Parameters of Traffic Simulation Scenario

The center frequency $f_{c}$ of the radar carrier was 77 GHz, the sampling rate $F_{s}$ of the radar was 50 MHz, the frequency modulation of the radar was 16.7 μs wide, the bandwidth was 500 MHz, and the transmission power was 25 dBm. The antenna was set to single-shot and multi-receiver, the receiving antenna was a line array, the number of channels was 6, and there were 128 distance-dimension fast Fourier transform (FFT) processing points and 128 speed-dimension FFT processing points. The spatial positions of the two stationary strong scattering targets (cars) are illustrated in Figure 15. In this figure, the radar vehicles are blue, the stationary vehicles are yellow and orange, and the stationary vehicles are 37 and 44 m away from the radar vehicles. The parameters of ground clutter, p and q, are 3 and 4, respectively. The radar has speeds of 10, 20, and 30 m/s.

#### 4.1.2. Simulation Results

The simulation was carried out using the above ground clutter modeling method. The spectrum formed by the range and relative velocity (i.e., the Range Doppler (RD) spectrum) of the ground clutter at different speeds of the radar is shown in Figure 16. It can be seen from this figure that due to the motion speed of the radar carrier, the ground clutter spectrum migrates from the zero-intermediate frequency to the radial speed of the radar carrier. At the same time, since there are two stationary vehicles in the scene, the radar reflection intensity of a vehicle is higher than the road reflection intensity, so two strong scattering bright spots appear at 37 and 44 m on the spectrum.

From the ground clutter modeling simulation, the position of the stationary vehicle in the RD spectrum of the ground clutter was determined, as shown in Figure 17.

Using the classical processing algorithm of the actual vehicle intelligent driving millimeter wave radar on the original simulation results of the ground clutter RD spectrum, we successively carried out moving target detection (MTD) processing, constant false-alarm rate (CFAR) detection and processing, and digital beam forming (DBF) processing.

The original simulation results for the ground clutter RD spectrum at different relative speeds after MTD processing are shown in Figure 18. It can be seen from this figure that the simulation results for the ground clutter RD spectrum after MTD processing not only reflected the real-time distribution characteristics of the ground clutter simulation scene but also introduced inherent frequency processing errors such as spectrum hollowing, spectrum broadening, spectrum shifting, and so on. Moreover, after MTD algorithm processing, the ground clutter simulation data did not cause additional unreasonable frequency interference components to the RD spectrum. Thus, for the MTD processing link, the theoretical verification effect of the ground clutter modeling method is good.

The simulation results of the ground clutter RD spectrum after MTD processing also need to be processed using radar CFAR detection. The simulation results after radar CFAR detection processing at relative speeds of 10, 20, and 30 m/s are shown in Figure 19.

It can be seen from this figure that the simulation results after CFAR detection and processing not only reflected the real-time range characteristics and speed characteristics of each target in the ground clutter simulation scene but also introduced the false alarm target, range error, and speed error associated with the real CFAR detection and processing. After CFAR detection and processing, the ground clutter simulation data did not cause additional unreasonable redundant false target interference to the radar target detection. Therefore, the ground clutter modeling method exhibits a good theoretical verification effect for CFAR detection and processing.

The simulation results of radar detection after CFAR detection processing also need to be processed by radar angle DBF. The simulation results after radar angle DBF processing at relative speeds of 10, 20, and 30 m/s are shown in Figure 20.

It can be seen from this figure that the simulation results after DBF processing not only reflected the real-time relative angle characteristics of each target in the ground clutter simulation scene but also introduced the angle error of the false alarm target associated with the actual DBF processing. Moreover, after DBF processing, the ground clutter simulation data did not add unreasonable redundant false target interference to the radar target detection. Therefore, the theory of ground clutter modeling was proven to be effective for DBF processing.

In addition, considering the direct influence of the relative amplitude of ground clutter on the radar detection results, we selected the rectangular window processing function, Hanning window processing function, and Hemingway window processing function, all of which are commonly used in actual radar, and applied the above ground clutter modeling method and the ground clutter simulation data generated by the simulation scene in order to calculate and plot the influence curve of the ground clutter amplitude on the number of radar detection targets, as shown in Figure 21.

As can be seen from this figure, the simulation results after rectangular window function processing indicate that the influence curve of the ground clutter amplitude on the number of radar detection targets was relatively weak, and the number of detection false alarm targets was small. On the other hand, the simulation results after Hanning window function processing and Hemingway window function processing reveal that the influence curve of the ground clutter amplitude on the number of radar detection targets was relatively strong, and there were more detected false alarm targets. The above characteristics are consistent with the processing results of the window function in actual radar detection. Therefore, the theoretical validation of ground clutter modeling method for the relationship between the amplitude of ground clutter and the number of targets detected by radar is good.

Overall, the ground clutter simulation data exhibited good consistency with actual radar in terms of radar RD spectrum generation, MTD processing, CFAR processing, DBF processing, and detection under different time domain window functions.

### 4.2. Simulation Application of Weather Clutter

In order to verify the effectiveness of the modeling method on weather clutter, we set up a traffic simulation scenario in which we could conduct simulation application experiments of weather clutter. We realized, of course, that weather clutter and ground clutter are usually present at the same time. Therefore, in the simulation scenario described below, the ground clutter simulation data were also generated synchronously. We distinguished the simulation scenes of highways with heavy rain, heavy snow, and dense fog. The simulation application test of weather clutter was performed separately.

The highway scene with heavy rain is shown in Figure 22.

The vehicle parameter settings in the highway scene with heavy rain are listed in Table 2.

The radar parameter settings in the highway scene with heavy rain are listed in Table 3.

The road surface parameters in the highway scene with heavy rain are listed in Table 4.

The weather parameters in the highway scene with heavy rain are listed in Table 5.

In the high-speed road scene with heavy rain, the synthetic RD spectrum of the target echoes and environmental clutter received by the radar are shown in Figure 23.

The target output after radar processing is shown in Figure 24.

It can be seen from this figure that in the highway scene with heavy rain, both the No. 3 car and the No. 4 car were normally detected by the radar. Due to the combined effects of roads and heavy rainfall, neither the No. 2 car nor the No. 5 car was detected normally, causing missing radar reports. At the same time, there were many false targets in the radar output, and there were certain distance measurement errors and speed measurement errors for the target, which were related to environmental clutter interference. The simulation results were in good agreement with the actual radar processing results.

We constructed the simulation test scenario for the autonomous emergency braking (AEB) system, as shown in Figure 25, and used MATLAB and PanoSim software to generate the radar environment simulation data using the aforementioned environmental clutter modeling method. We then tested the intelligent driving AEB decision algorithm, as shown in Figure 26.

The test results are shown in Figure 27 and Figure 28 below, in which 1 indicates that the AEB system has not started, 2 indicates that the AEB system is in the warning state, and 3 indicates that the AEB system is in the braking state. The test results of the AEB decision algorithm without the support of the radar environmental clutter simulation data were then compared and analyzed.

From the comparison results, we can see that the radar environmental clutter simulation method presented in this study exhibited obvious advantages. When there was no clutter in the simulation data, the test results of the AEB decision algorithm were idealized, the transition of the AEB system working state was stable, and it could not reflect the dynamic changes that occur during an actual road test. After the radar clutter simulation method was added, however, the potential defects and shortcomings of the AEB algorithm were clearly exposed, and the AEB state in the test results was closer to the actual road test results. Compared with other test methods, the simulation method of radar environmental clutter can support the repeatability test of the decision algorithm more effectively and improve the optimization of the decision algorithm.

## 5. Discussion and Conclusions

This study proposed a simulation method of the millimeter wave radar virtual test environment for intelligent driving. The work and innovations of this project are summarized as follows:According to the characteristics of intelligent driving millimeter wave radar, and based on the principles of statistics and electromagnetism, millimeter wave radar for intelligent driving was analyzed for the first time, and the mechanism of environmental clutter was examined in detail.Based on the surface characteristics in intelligent vehicle traffic scenes, the surface differences between highway traffic roads, urban traffic roads, and rural traffic roads were investigated. A simulation method for the ground clutter in millimeter wave radar for intelligent driving was proposed.In terms of the weather characteristics of various intelligent vehicle traffic scenes, an analysis of the statistical distribution characteristics of rain, snow, and fog provided us with the capability to distinguish weather particle size, inter-particle density, wind speed, and wind direction. A simulation method for rain, snow, and fog weather clutter of millimeter wave radar for intelligent driving was proposed.The surface distribution characteristics under typical scenes were designed. Based on the signal processing and data processing algorithms of millimeter wave radar, the effectiveness of the ground clutter simulation method was verified.The distribution characteristics of rain, snow, and fog in typical scenes were designed. Based on the radar signal processing and data processing algorithms, the effectiveness of the weather clutter simulation method was verified.The research content of this study is an important part of the simulation model of intelligent vehicle millimeter wave radar, since it supplies the missing environmental clutter modeling and simulation method in the simulation model of intelligent vehicle millimeter wave radar, thereby solving a key problem and shortcoming.

The research results and conclusions of this study are significant to the field of intelligent driving simulation testing based on millimeter wave radar.

## Figures and Tables

**Figure 1 sensors-20-01929-f001:**
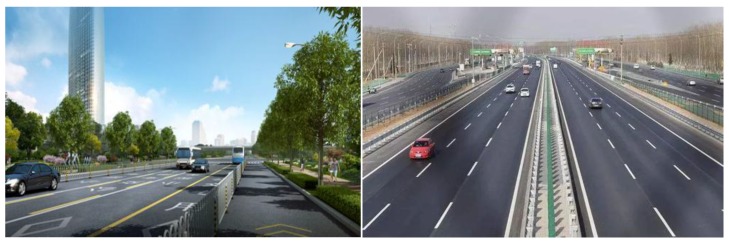
Surface traffic scene.

**Figure 2 sensors-20-01929-f002:**
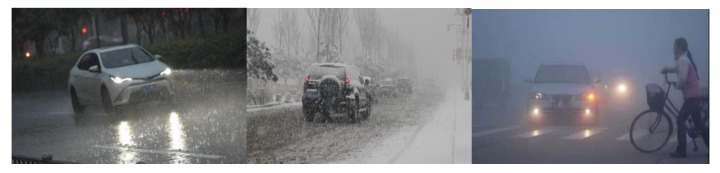
Traffic scene on rainy, snowy, and foggy days.

**Figure 3 sensors-20-01929-f003:**
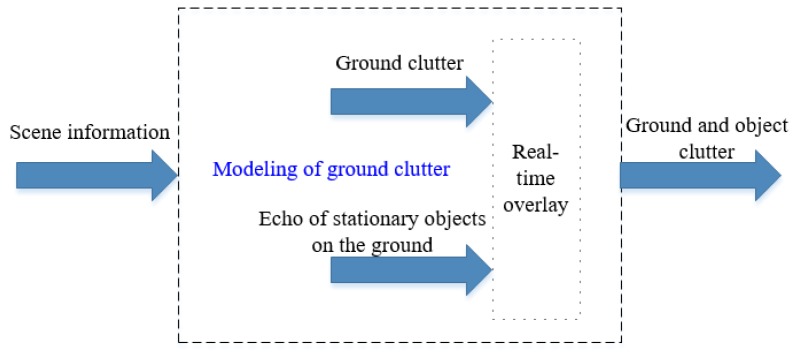
Architecture of ground clutter modeling.

**Figure 4 sensors-20-01929-f004:**
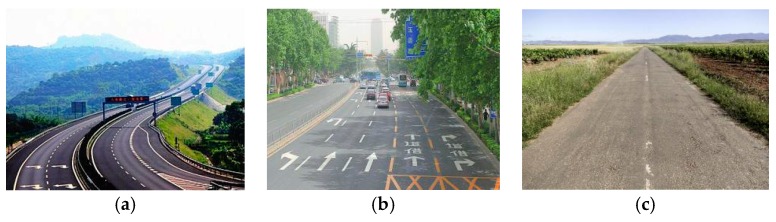
Road environment for three types of traffic: (**a**) Highways; (**b**) Urban roads; (**c**) Rural roads.

**Figure 5 sensors-20-01929-f005:**
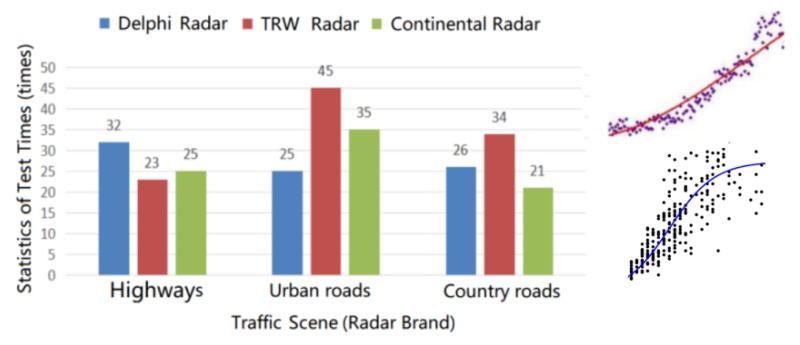
Analysis of ground clutter data fitting.

**Figure 6 sensors-20-01929-f006:**
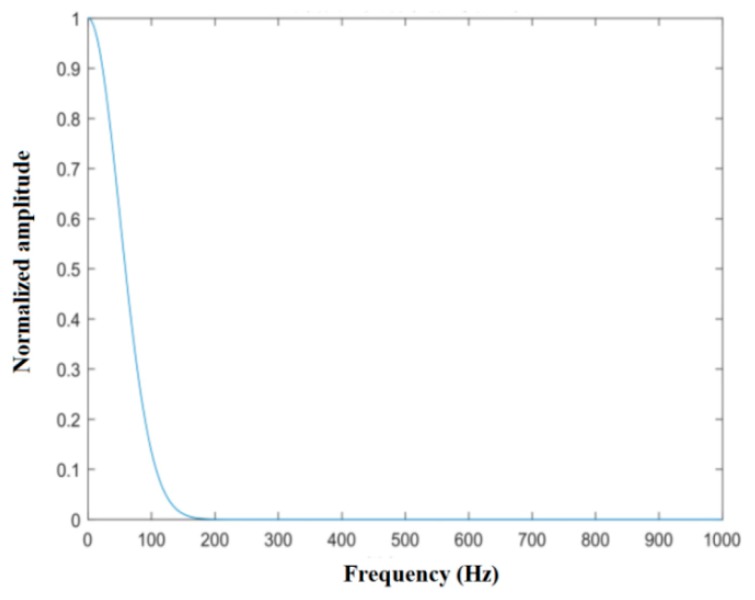
Spectral density curve of the Gaussian power spectrum.

**Figure 7 sensors-20-01929-f007:**
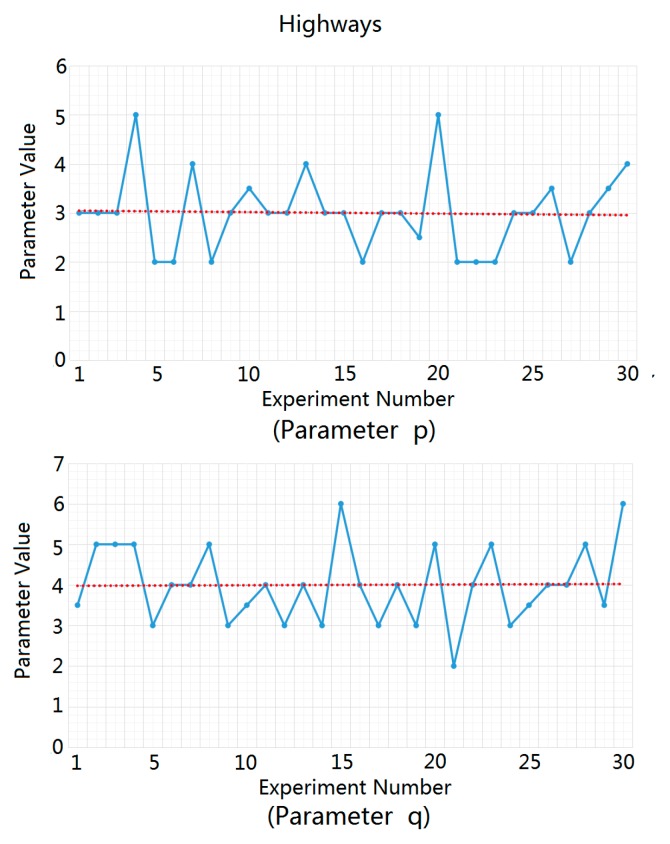
Solution process of parameters p and q for highways.

**Figure 8 sensors-20-01929-f008:**
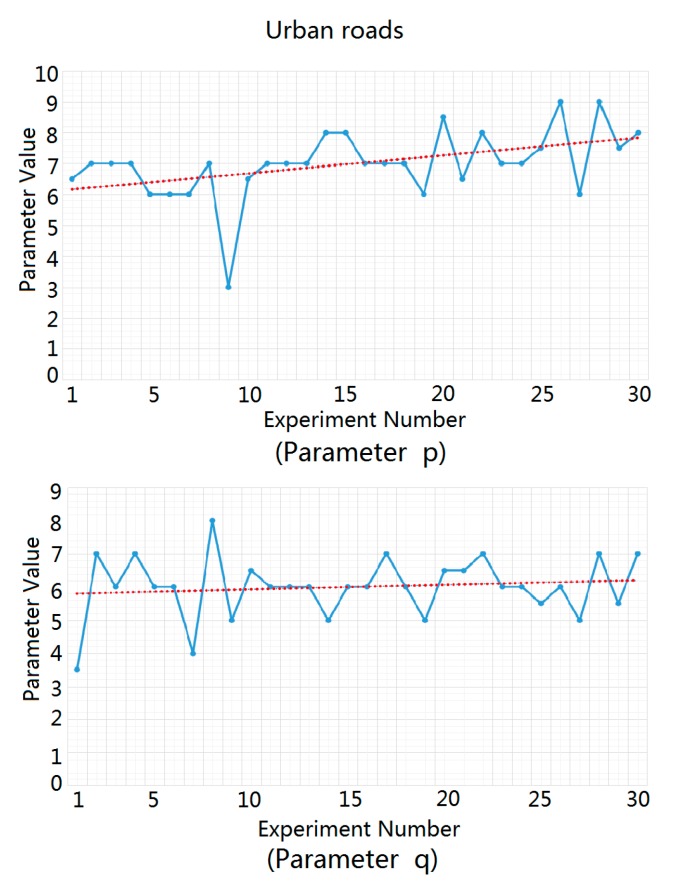
Solution process of parameters p and q for urban roads.

**Figure 9 sensors-20-01929-f009:**
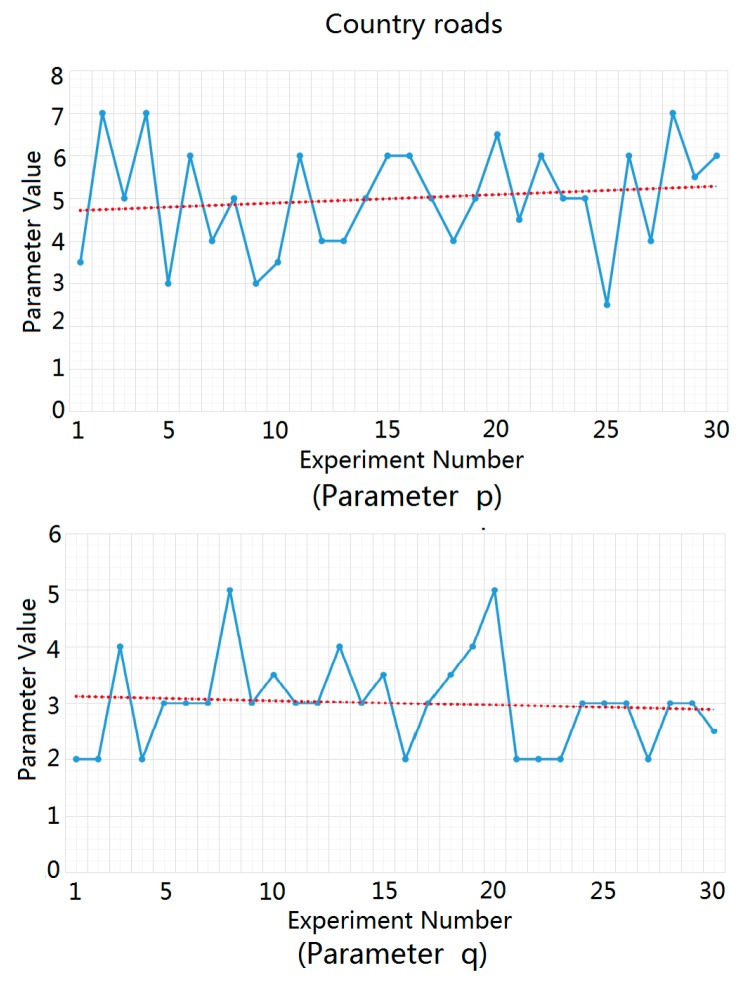
Solution process of parameters p and q for rural roads.

**Figure 10 sensors-20-01929-f010:**
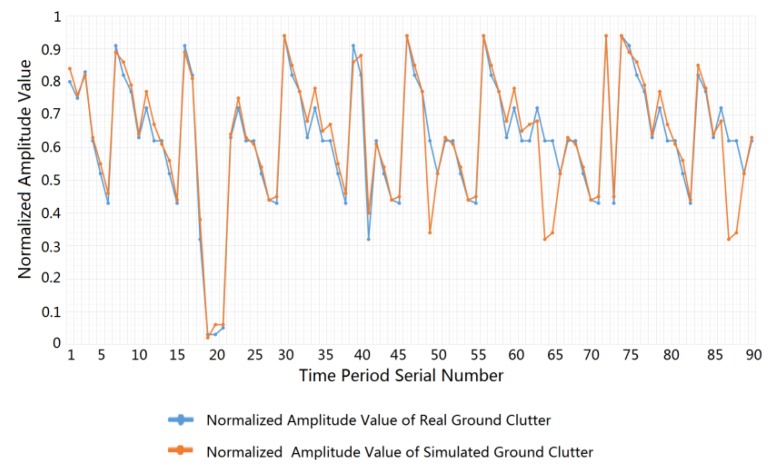
Comparison of amplitude values between actual clutter and simulated clutter.

**Figure 11 sensors-20-01929-f011:**
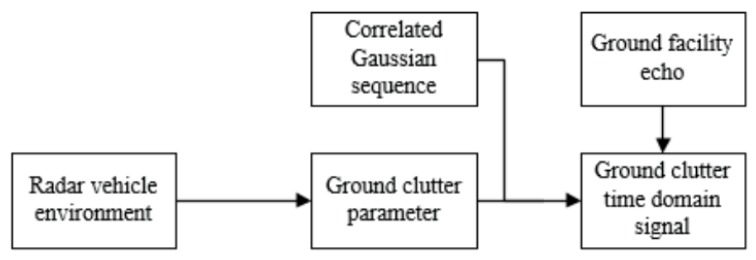
Simulated flow of ground clutter.

**Figure 12 sensors-20-01929-f012:**
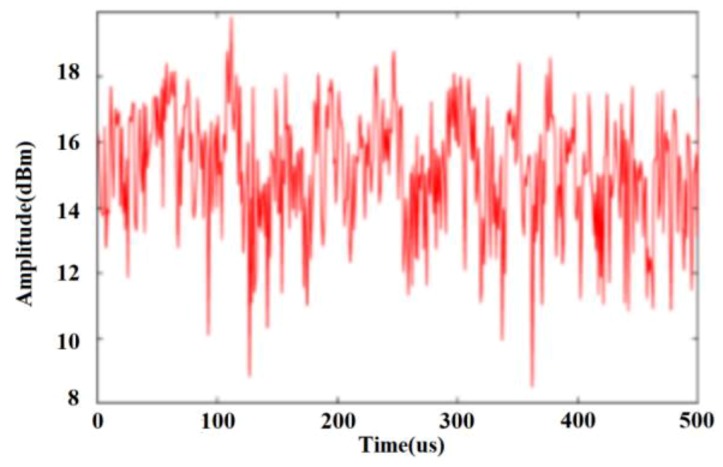
Ground clutter modeling method used to generate the Weibull distribution time domain signal.

**Figure 13 sensors-20-01929-f013:**
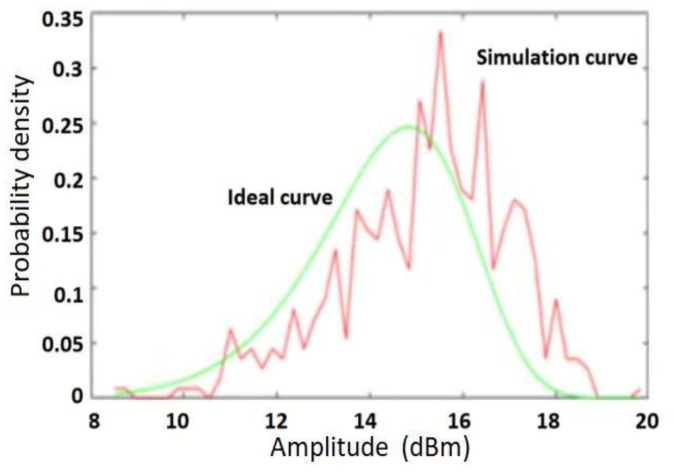
Comparison of complex Weibull clutter and ideal Weibull curve generated by the ground clutter modeling method.

**Figure 14 sensors-20-01929-f014:**
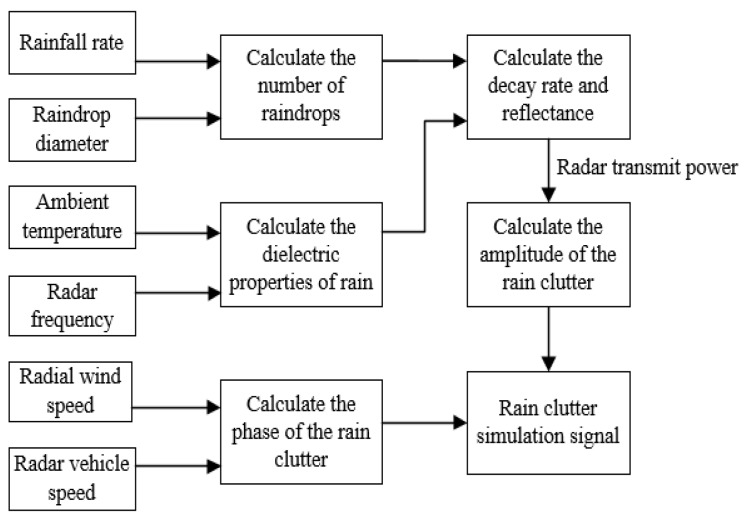
Rain clutter modeling method.

**Figure 15 sensors-20-01929-f015:**
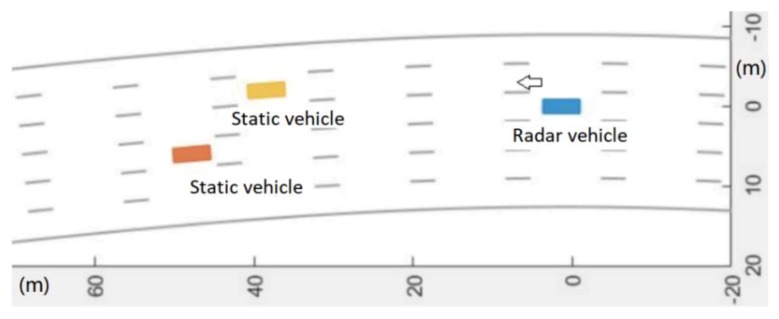
Schematic diagram of ground clutter scene.

**Figure 16 sensors-20-01929-f016:**
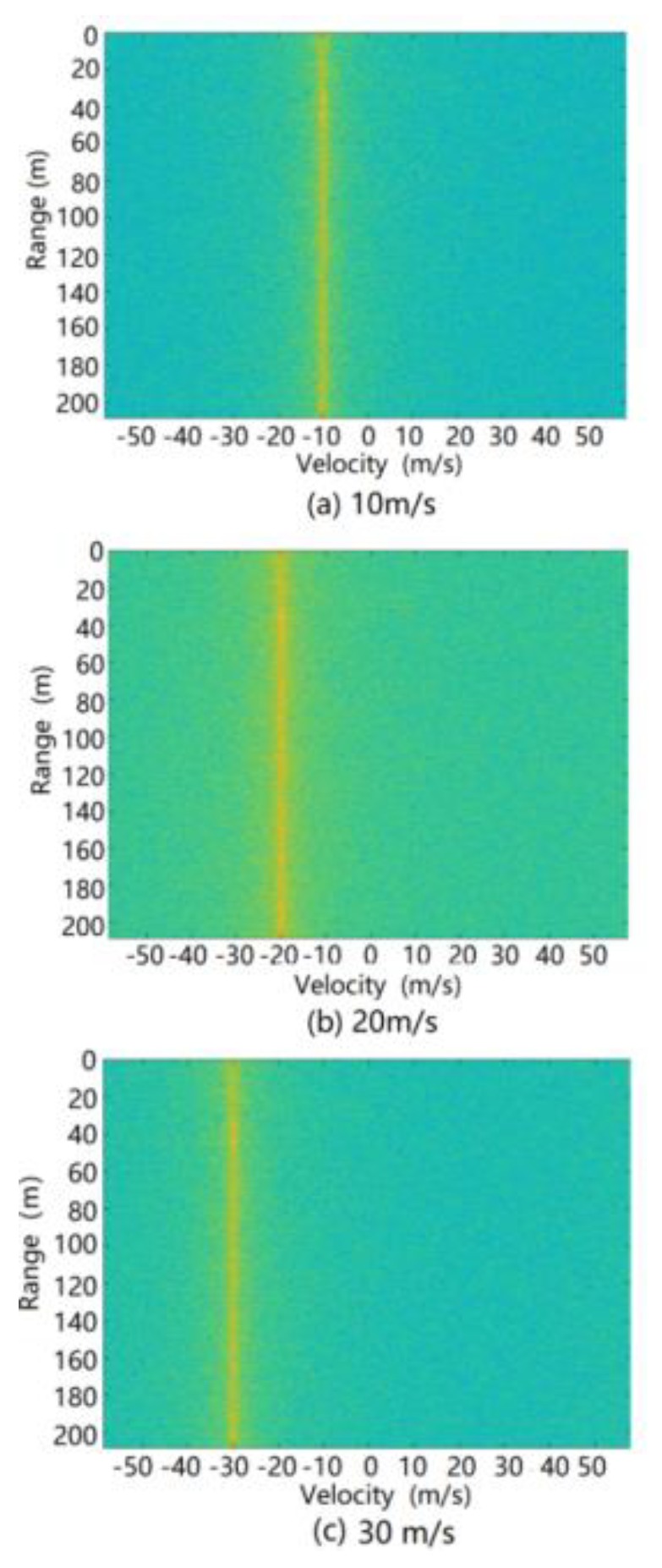
RD spectrum of simulated ground clutter at different speeds of radar-carrying vehicles.

**Figure 17 sensors-20-01929-f017:**
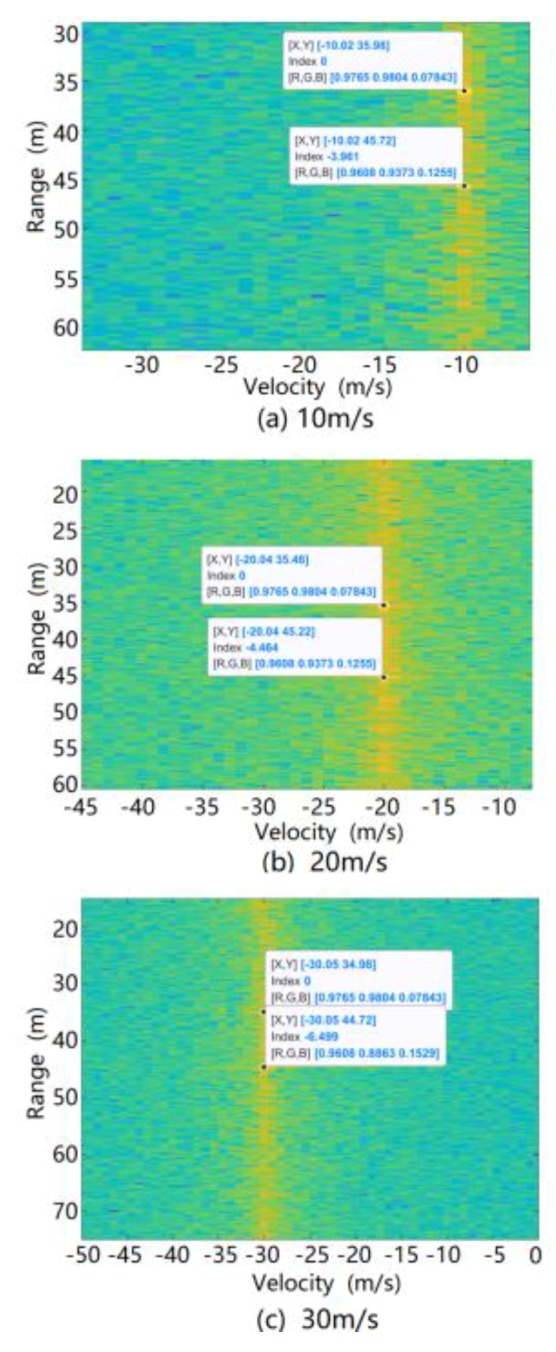
Position of stationary vehicle in RD spectrum of ground clutter during simulation.

**Figure 18 sensors-20-01929-f018:**
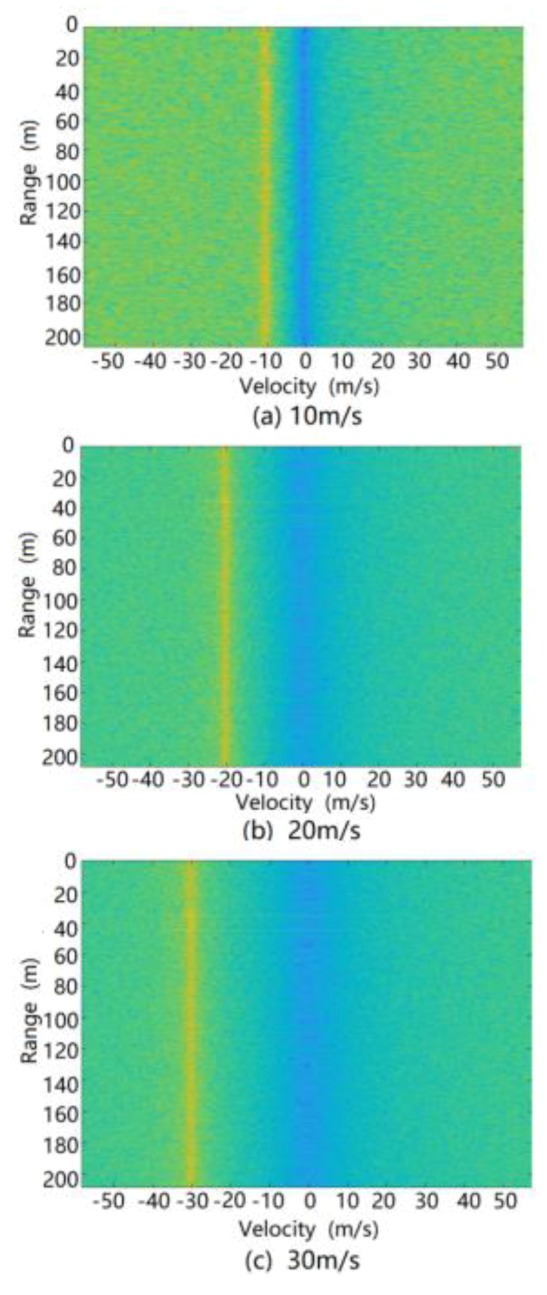
RD spectrum of ground clutter treated by moving target detection (MTD) processing during simulation.

**Figure 19 sensors-20-01929-f019:**
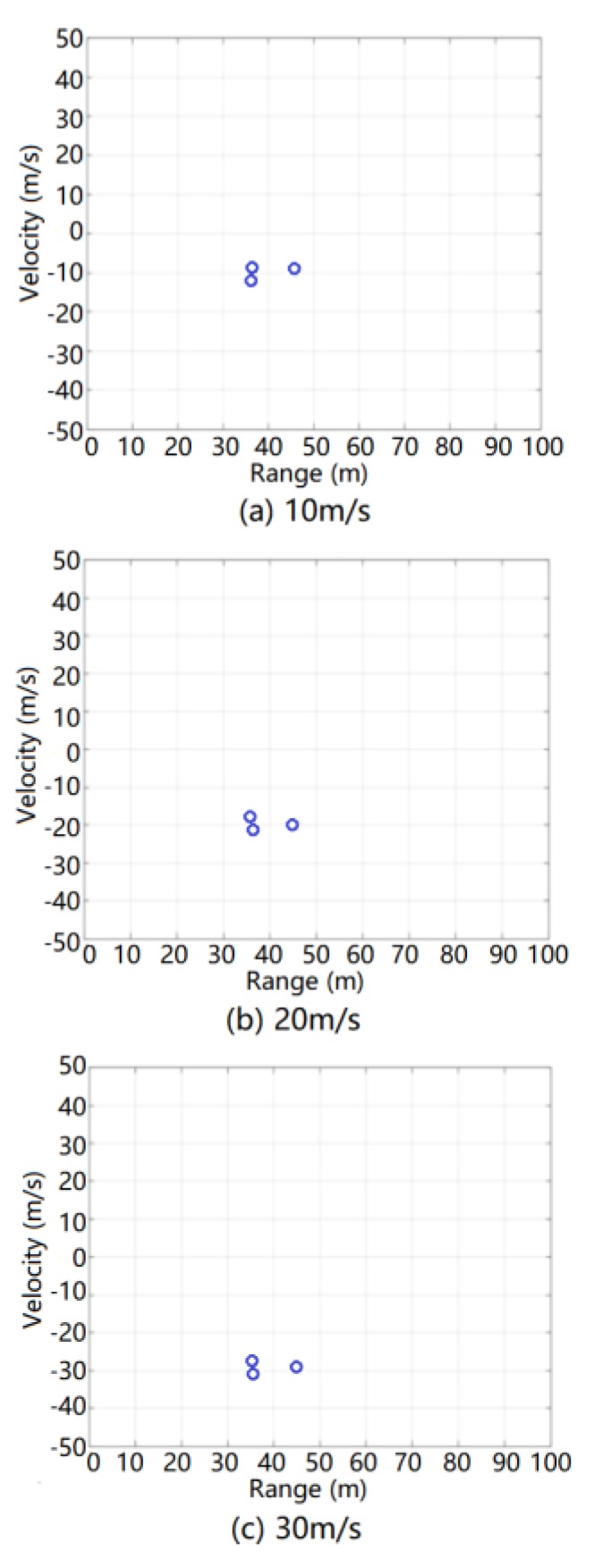
Results of ground clutter after constant false-alarm rate (CFAR) detection and processing during simulation.

**Figure 20 sensors-20-01929-f020:**
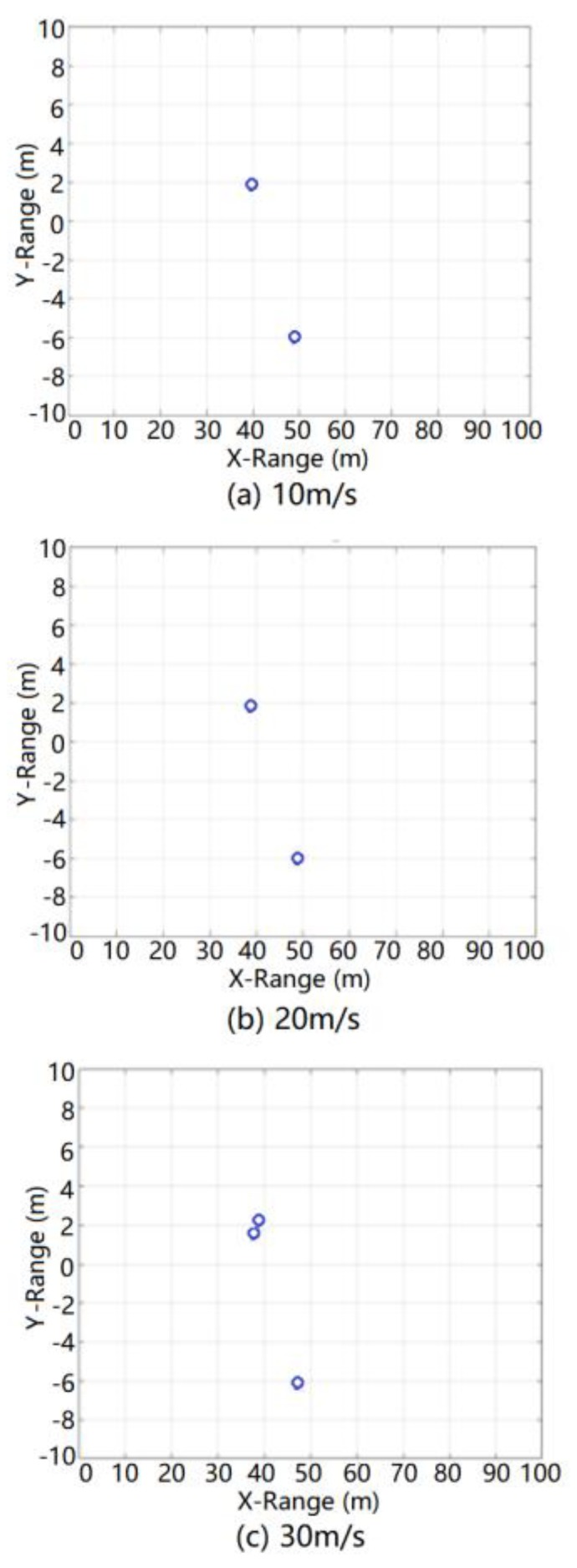
Results of ground clutter after digital beam forming (DBF) processing during simulation.

**Figure 21 sensors-20-01929-f021:**
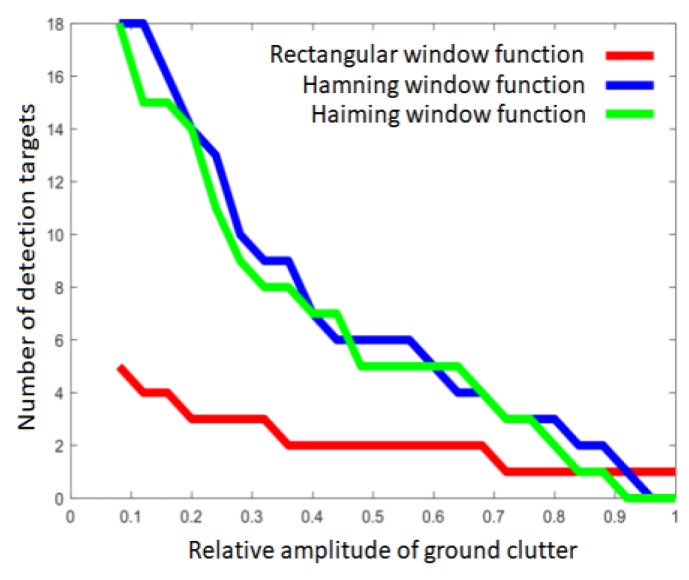
Effect of relative amplitude of surface clutter on test results during simulation.

**Figure 22 sensors-20-01929-f022:**
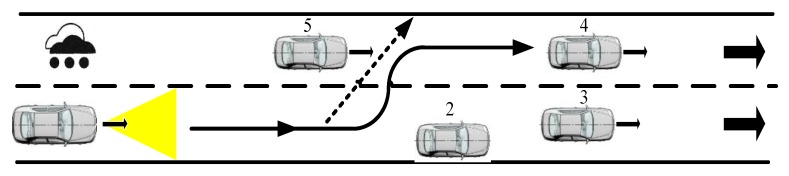
Highway scene with heavy rain.

**Figure 23 sensors-20-01929-f023:**
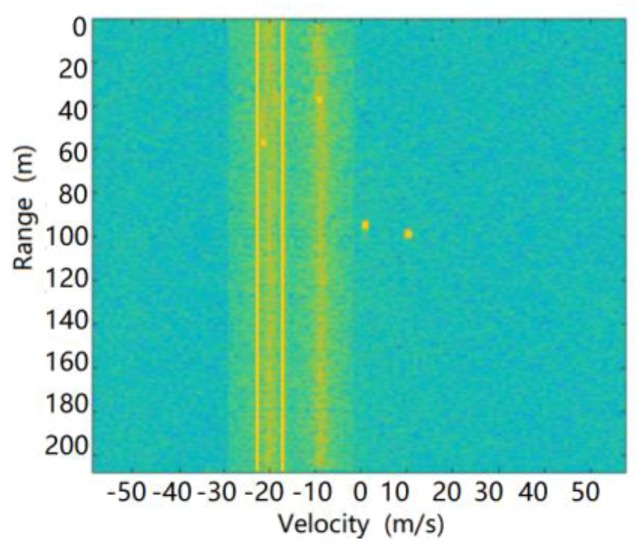
RD spectrum of radar echoes in highway scene with heavy rain.

**Figure 24 sensors-20-01929-f024:**
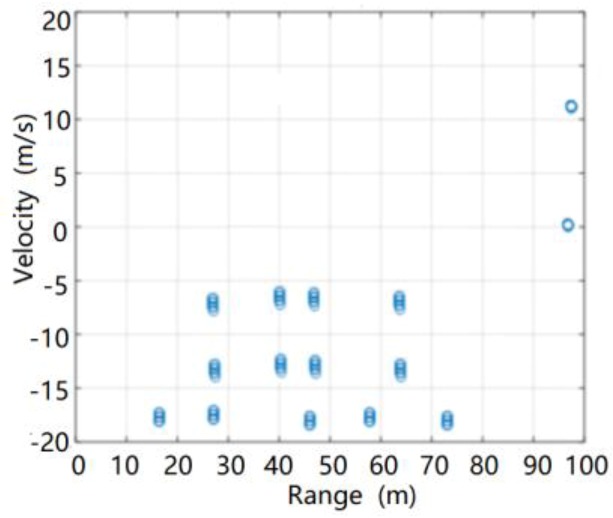
Target output of radar in highway scene with heavy rain.

**Figure 25 sensors-20-01929-f025:**
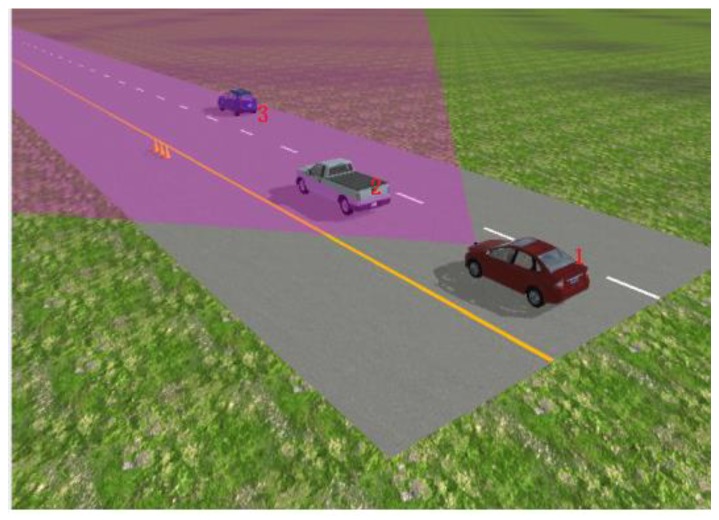
Test scenario for autonomous emergency braking (AEB) decision algorithm.

**Figure 26 sensors-20-01929-f026:**
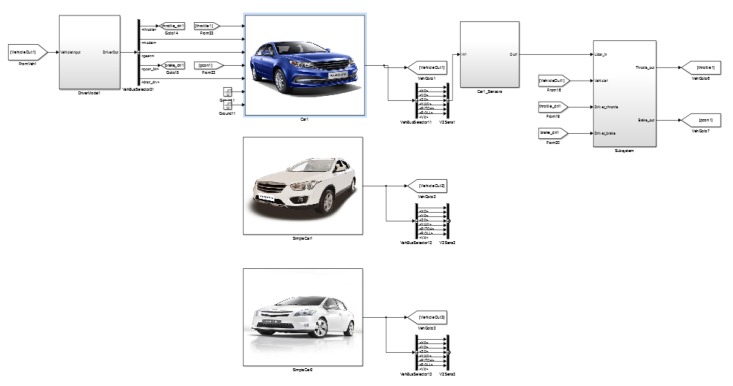
Simulink model file of AEB decision algorithm based on MATLAB and PanoSim software.

**Figure 27 sensors-20-01929-f027:**
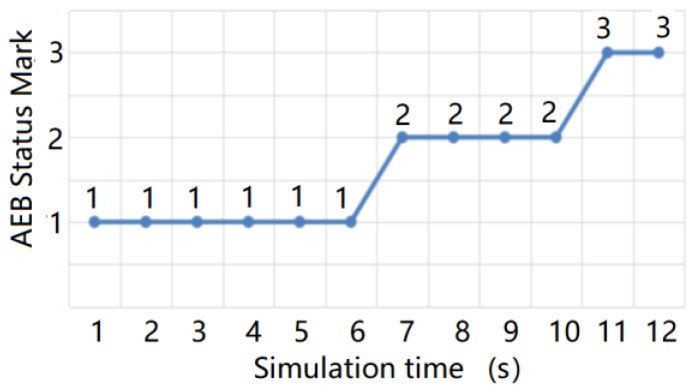
AEB state change without radar environmental clutter simulation data.

**Figure 28 sensors-20-01929-f028:**
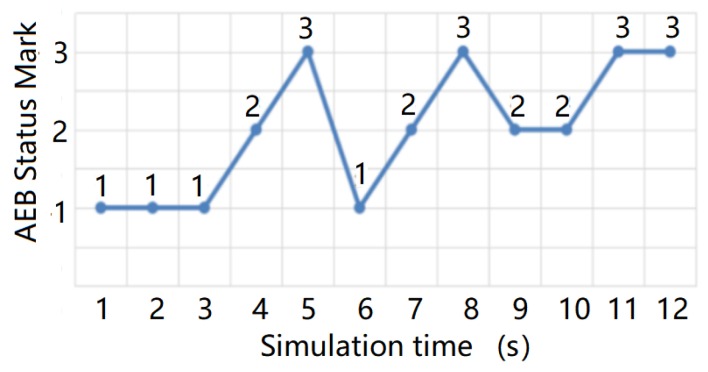
AEB state change with radar environmental clutter simulation data.

**Table 1 sensors-20-01929-t001:** Recommended values of ground clutter parameters.

Parameter	Highway	Urban Road	Rural Road
*p*	3	7	5
*q*	4	6	3

**Table 2 sensors-20-01929-t002:** Vehicle information in highway scene with heavy rain.

Parameter	Radar Car	Car Number 2	Car Number 3	Car Number 4	Car Number 5
Driving speed	20 m/s	0 m/s	20 m/s	30 m/s	10 m/s
Relative distance	0 m	60 m	100 m	100 m	40 m

**Table 3 sensors-20-01929-t003:** Radar parameter settings in highway scene with heavy rain.

Parameter	Parameter Value
Radar center carrier frequency	77 GHz
Transmission power	25 dbm
Transmitting antenna gain	27 dB
Receiver antenna gain	27 dB

**Table 4 sensors-20-01929-t004:** Pavement parameters in highway scene with heavy rain.

Parameter	Parameter Value
Pavement parameter p	3
Pavement parameter q	4

**Table 5 sensors-20-01929-t005:** Weather parameters in highway scene with heavy rain.

Parameter	Parameter Value
Rainfall rate	10 mm/h
Raindrop diameter	2 mm
Ambient temperature	27 °C
Wind speed	10 m/s

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
