# Peer review of "Research on a Simulation Method of the Millimeter Wave Radar Virtual Test Environment for Intelligent Driving"

_sensors, 2020, doi:10.3390/s20071929_

Round 1

Reviewer 1 Report

This is a good paper that opens a line of research, but needs housing.

With which software was the traffic simulation created? Have the traffic data been calibrated? Have the radar data been calibrated with the real data? Add a paragraph on simulation data calibrations. With many types of radar simulations have we done? Are the analyzes described in the document valid for all radars? The text in figure 11 is not very visible The text in figure 12 is not very visible The text in figure 13 is not very visible The text in figure 14 is not very visible The text in figure 15 is not very visible

Author Response

Dear Reviewer,

First of all, we would like to express our sincere gratitude for giving us a chance to revise our manuscript. Your comments and suggestions are very valuable and helped us to improve the manuscript quality.

After carefully studying your comments, we have thoroughly revised the manuscript. A point-by-point summary of the revisions is given below.

Reviewer 2 Report

    To address the virtual test of intelligent driving and study the key problems in modeling and simulating millimeter wave radar environmental clutter,the model and simulation method is proposed for environmental clutter of millimeter wave radar for intelligent driving. First, according to the characteristics of intelligent vehicle millimeter wave radar, the generation mechanism of radar environmental clutter are analyzed. Then the simulation method of radar clutter under environmental conditions such as surface, rainfall, snowfall, and fog are deduced and designed.

  The idea of this article is clear and the method is feasible in theory.However, this article still has the following deficiencies.

(1)The introduction does not elaborate the relevant research on the simulation method of millimeter wave radar  virtual test environment by domestic and foreign scholars at present, which needs to be supplemented. (2) In section 3.1, the author establishes the the model of ground clutter by considering the scattered echo of the ground clutter and the stationary objects on the ground. Through a large number of measured data, it is known that the power spectrum of the ground clutter approximates a Gaussian distribution. But the author has not explained how much the error between Gaussian distribution and real data distribution is . Similarly, the amplitude distribution of ground clutter approximates a Weibull distribution, and the error between Weibull distribution and the amplitude of real ground clutter needs to be explained. (3)Please explain how the values of the parameters in table 1 are obtained, and please mark the references if quoting. (4)In fig. 13, the author said that the processing result is in good agreement with the actual radar processing result, and there is no comparative experiment to support the conclution. (5)Figs. 14, 15 and 16 lack comparative experiments with actual radar detection results. (6)The author of the whole experiment only said that the processing result is the same as the actual radar processing result, and there is no comparative data as support and lacking of the same basis for judging. (7)What are the advantages of this modeling method, and what are its advantages compared with other modeling methods, need to be explained in the introduction and experiments.      

Author Response

Dear Reviewers,

First of all, we would like to express our sincere gratitude for giving us a chance to revise our manuscript. Your comments and suggestions are very valuable and helped us to improve the manuscript quality.

After carefully studying your comments, we have thoroughly revised the manuscript. A point-by-point summary of the revisions is given below.

Round 2

Reviewer 1 Report

The authors have expanded and arranged the paper correctly.